# Effects of Policy for Controlling Agricultural Non-Point Source Pollution in China: From a Perspective of Regional and Policy Measures Differences

**DOI:** 10.3390/ijerph20043741

**Published:** 2023-02-20

**Authors:** Chunlin Hua, Jiuhong Zhang, Zhiru Long, Richard T. Woodward

**Affiliations:** 1College of Economics and Management, Southwest University of Science and Technology, Mianyang 621010, China; 2College of Economics and Management, China Agricultural University, Beijing 100083, China; 3School of Economic, The University of Sydney, Sydney, NSW 2006, Australia; 4Department of Agricultural Economics, Texas A&M University, College Station, TX 77843-2124, USA

**Keywords:** Agricultural Non-Point Source pollution, policy strength, policy measures, regional difference, dynamic panel data model

## Abstract

The concerns about the contradiction between agricultural production and Agricultural Non-Point (ANPS) pollution has become increased with economic development in China. Government interventions are key to controlling ANPS pollution through the establishment of laws and policies. This paper uses the entropy method to calculate the emissions amount of ANPS pollution and policy strength of 31 provinces from 2010 to 2019 in China. The dynamic panel data model with system generalized moment is used to estimate the impacts of policies with different measures on ANPS pollution emission. According to our findings, China’s policies have been helpful in controlling ANPS pollution though there are important regional differences. Moreover, four types of policy measures all contribute to the reduction in ANPS pollution. These findings improve our understanding of the relationship between policies and ANPS pollution in the analyzed period, thus providing support for the formulation of pollution management strategies in the next stage.

## 1. Introduction

Agricultural Non-Point Source (ANPS) pollution is pollution caused by solid or dissolved pollutants in agricultural production and rural life. ANPS pollution has become a worldwide water pollution problem [1,2], and developing countries account for 74% of ANPS pollution [3,4]. The results of China’s second national pollutant source survey in 2017 showed that, among the various pollution sources including industry, agriculture, and life, the chemical oxygen demand emissions from the agricultural sector accounted for 49.77% (10.67 million tons), total nitrogen emissions accounted for 46.52% (0.22 million tons), and total phosphorus emissions accounted for 67.22% (0.21 million tons). Despite increases in the number of monitoring points and improvements in national statistics, the measurement of ANPS emissions is difficult and imperfect. Hence, the magnitude of ANPS pollution emissions may be even greater.

The market equilibrium level of ANPS pollution will be greater than the socially optimal because it is a classic externality; the public does not receive any compensation for the pollution damages caused by ANPS pollution. Therefore, government interventions are key to controlling ANPS pollution through the establishment of laws and policies, such as setting targets for total ANPS pollution, monitoring emissions and pollution loads, and setting discharge standards and technical specifications for ANPS pollutants [5,6,7,8]. Government policies are more effective in China than in Western countries [9] and the Chinese central government is increasingly paying attention to ANPS pollution.

More policies have focused on controlling ANPS pollution, including the “Zero Growth Action Plan for Fertilizer Use by 2020” and the “Zero Growth Action Plan for Pesticide Use by 2020” [10]. With regard to ANPS, political decision-making, and national governance practices are said to have “functional commonality and consistency in content” [11] which means that the prevention and control of ANPS pollution are seen as a high priority, referred to as the “will of the state” [12,13,14]. After intensive efforts during the Thirteenth Five-Year Plan period (2016–2020), the prevention and control of ANPS pollution have achieved remarkable results. Compared with 2015, the number of agricultural chemical fertilizers and pesticides input decreased by 10.28% and 20.85% respectively in 2019. Moreover, the utilization rate (The utilization rate is the ratio of nutrients from chemical fertilizer absorbed by crops. The more fertilizer nutrients are absorbed the less ANPS pollution emissions will be which means higher utilization rate is better for the environment) of chemical fertilizers for China’s three major grain crops of rice, wheat, and corn in 2020 increased to 40.2%, an increase of 5 percentage points from 2015. The utilization rate of chemical pesticides is 40.6%, an increase of 4 percentage points over 2015 [15]. The reduction of chemical fertilizer input and the increase in utilization rate suggests that green agriculture policies have been effective. The effectiveness of policies in China is supported by the analysis of Cao et al. (2014) [16] who found that the ecological agriculture policy has increased the efficiency of agricultural production, reduced the number of chemical fertilizers and pesticides, and improved water quality in the Erhai Lake Basin from 2000 to 2012; Zhang et al. (2017) [17] found that the policies of environmental management can achieve better effects on water quality in a rural area in the short term.

Nonetheless, weaknesses have been pointed out in the suite of current ANPS policies. The top-level framework of China’s ANPS pollution control policy has been basically established, but the goals of existing national, provincial, and city policies are inconsistent at different levels [18], lacking a clear legal basis, a strict supervision mechanism [19], a clear application boundary [20,21], and the basic information on ANPS pollution when formulating a policy [22]. It is difficult for Chinese farmers to understand policy targets and reduce the chemical fertilizer inputs in a short period, thus the cost of implementing policies is relatively high [23,24]. Shi et al. (2020) [25] found that ANPS policies could have a strong impact on farmers’ behavior but if the policies affect only a few farmers, ultimately they will achieve little. Agricultural emissions continue to account for about half of the total water pollution emissions in China [26]. There are, however, significant regional differences in agricultural production in China which lead to different policy effects. For example, the proportion of ANPS pollution of cereal crops in the northeast, of vegetable crops in southeastern coastal areas is higher than in other areas in China [27]. Xu and Xue (2019) [28] argue that urbanization will have the effect of intensifying ANPS pollution in the northeast and reducing it in the east and west. Although research on controlling ANPS pollution in China has grown rapidly since 1990 [29], since agriculture is the main industry in China, more research is needed.

To the best of our knowledge, there are relatively few studies that systematically analyze the policies being used to control ANPS pollution and the effectiveness of policies for controlling ANPS pollution has not been quantitatively evaluated with consideration of regional differences. The first contribution of this paper is that we quantify the strength of policies implemented from 2010 to 2019 in 31 Chinese provinces. Secondly, we examine how well ANPS pollution is controlled by different types of measures among the policies, including administrative regulations, economic incentives, technical support, and educational efforts. Thirdly, we investigate how policies affect ANPS pollution emissions in different geographic areas.

Using official data and policy documents, we calculate the emissions of ANPS pollution and policy strength using the entropy method. Then, a dynamic panel data model is built to estimate the effects of policies on ANPS pollution with further discussion about the effects of different policy measures. Next, we expand our study to estimate the policy effects with regional differences. Finally, we discuss our findings and present the conclusion and policy implications.

## 2. Data Source

### 2.1. Data for Calculation of ANPS Pollution Emissions

This paper considers the ANPS pollution emissions from chemical fertilizers and crop residues in the agricultural planting process, the waste loading from the pig, cattle, poultry, and aquaculture in the animal breeding industry, and the discharge of rural domestic sewage. The data are derived from the China Statistical Yearbook, China Rural Statistical Yearbook, and provincial statistical yearbooks from 2010 to 2019. The loading coefficients of Nitrogen (N), Phosphorus (P), and chemical oxygen demand (COD) are the proportion of pollutants runoff into water per unit of different sources and derived from the Manual of “The First National Survey of Pollution Sources—Manual of Fertilizer loading coefficients” in China 2010.

### 2.2. Data for Calculation of Policy Strength

The data on ANPS pollution policies are obtained from the Peking University Law database which contains all the laws and regulations of China since 1949. This database is a legal information retrieval system jointly launched by the Legal Artificial Intelligence Laboratory of Peking University and Beijing Beida Yinghua Technology Co., LTD. Since our goal is to estimate the effectiveness of policies in controlling ANPS, we used the following seven keywords when searching the law database: “Agricultural Non-Point Source pollution”, “agricultural environment”, “chemical fertilizer”, “livestock and poultry breeding”, “aquaculture”, “crop residue”, and “rural domestic sewage” (Keywords in Chinese are: “农业面源污染”, “农业环境”, “化肥”, “畜禽养殖”, “水产养殖”, “秸秆”, “农村生活污水”). From the policies identified, we selected policies promulgated and implemented by related government departments and the legislature at the provincial level and remove “replies”, “approvals”, “Submission Documents” and other weakly normative and instructive regulatory documents, as well as documents that are relatively weakly related to ANPS pollution. This resulted in a policy dataset that consists of 1113 policy documents from 31 provinces (including municipalities directly under the central government and autonomous regions) from 2010 to 2019.

## 3. Methods

### 3.1. Calculation Equations of ANPS Pollution Emissions

We calculate the ANPS pollution emissions from crops, animal breeding, aquaculture, and rural community sources (shown in Table 1). Unabsorbed nutrients from chemical fertilizers used in agriculture leach into groundwater and surface water due to rain and irrigation, and this causes surface water eutrophication and groundwater nitrate pollution [30,31]. Incineration of farmland crop residue or crop residue dumping will cause organic matter and microorganisms to enter the water body and cause water pollution [32,33,34]. According to the second survey report of pollution sources in China in 2016, livestock and poultry manure ranked first among all pollution sources from the agricultural sector. The feces, urine, and sewage generated during the livestock and poultry breeding process lead to a large amount of loss of nitrogen and phosphorus entering waterways [35,36]. Aquaculture also results in pollution due to excessive inputs of bait, the production of excrement, and the use of chemicals and antibiotics [37]. Finally, with the improvement of the living standards in China’s rural areas, the discharge of domestic sewage is increasing which has become one of the main sources of water pollution [38]. All of these sources are accounted for in our analysis.

ANPS pollution can be calculated for different sources with differences in the geographic characteristics of pollutants [39,40,41]. The equations used to calculate the Total Nitrogen (TN), Total Phosphorus (TP), and COD emissions from agricultural production units in each province every year are provided in Table 2.

In each equation, the loading coefficient, λ, is for N, P, and COD for each pollutant (details of the loading coefficients for all sources are in the Appendix A). For fertilizer, aquaculture, and rural domestic sewage, the pollution is estimated by multiplying the variable from Table 1 by the respective loading coefficient. For example, the total amount of the kth fertilizer (1 = nitrogen fertilizer, 2 = phosphate fertilizer, 2 = compound fertilizer) input is $T_{k}$; the loading coefficient s of N and P in the kth fertilizer are $\lambda_{{\mathrm{kF}}_{n}}$ and $\lambda_{{\mathrm{kF}}_{p}}$. The emissions of nitrogen and phosphorus, $F_{n}$ and $F_{P}$, equal $T_{k}$ multiplies $T_{n}\times \lambda_{{\mathrm{kF}}_{n}}$ and $T_{p}\times \lambda_{{\mathrm{kF}}_{p}}$.

More coefficients need to be included for livestock poultry and crop residue. When calculating the emissions from livestock and poultry farming, $T_{z}$ is the end-of-period or slaughtering quantity of the zth livestock and poultry (1 = poultry, 2 = pigs, and 3-cattle) and $\theta_{\mathrm{zL}}$ is the growth cycle of the *z*th livestock and poultry. The growth cycle of $\theta_{\mathrm{zL}}$ is 140, 180, and 365 days for poultry, pigs, and cattle respectively. The loading coefficient s, $\lambda_{{\mathrm{zL}}_{n}},{\;\lambda }_{{\mathrm{zL}}_{p}},{\;\lambda }_{{\mathrm{zL}}_{n}}$, are multiplied by $T_{z}$ and $\theta_{\mathrm{zL}}$ to get $L_{n}$, $L_{p}$, $L_{\mathrm{cod}}$, the emissions of N, P, and COD from livestock and poultry farming. For crop residue, the yield of the mth crop is $T_{m}$; the crop residue production coefficient of the mth crop is $\varphi_{m}$; the loading coefficient s of N, P, and COD in the crop residue are $\lambda_{S_{n}}$, $\lambda_{S_{p}}$, $\lambda_{S_{c}}$; $S_{n}$, $S_{p}$, $S_{c}$ are the total amount of N, P, COD emissions from crop residue which come from $T_{m}$ multiplied by $\varphi_{k}$ and $\lambda_{S_{n}}$, $\lambda_{S_{p}}$, $\lambda_{S_{c}}$.

Finally, we add up the emissions from all pollution sources to obtain the ANPS pollution emissions of *i* = 1, …, 31 provinces (municipalities and autonomous regions) from *t* = 2010 to 2019 (Details of the calculation process are in the Appendix A):(1)$${\mathrm{ANPS}}_{it}={\mathrm{wTN}}_{\mathrm{it}}\left(F_{\mathrm{nit}}+S_{\mathrm{nit}}+L_{\mathrm{nit}}+E_{\mathrm{nit}}+A_{\mathrm{nit}}\right)+{\mathrm{wTP}}_{\mathrm{it}}\left(F_{\mathrm{pit}}+S_{\mathrm{pit}}+L_{\mathrm{pit}}+E_{\mathrm{pit}}+A_{\mathrm{pit}}\right)+{\mathrm{wCOD}}_{\mathrm{it}}\left(S_{\mathrm{CODit}}+L_{\mathrm{CODit}}+E_{\mathrm{CODit}}+A_{\mathrm{CODit}}\right)$$
where ${\mathrm{ANPS}}_{it}$ is the ANPS pollution emissions in the *i*th province in the *t*th year; the weights of TN, TP, and COD emissions, ${\mathrm{wTN}}_{i,t}$, ${\mathrm{wTP}}_{i,t}$, and ${\mathrm{wCOD}}_{i,t}$, are calculated by the entropy method [42,43] (Entropy method is one of the common methods to determine weight and is a relatively objective and widely used than the analytic hierarchy process and coefficient of variation method. This paper draws on the improvement of entropy method made by Yang and Sun (2015), and adds time variable for analysis, so as to realize the comparison between different years.). It is worth emphasizing that we consider the differences in topography, climate, farming methods, crops, or breeding types of each province (municipalities and autonomous regions), and we use different loading coefficients for each area. However, the loading coefficients are held constant. Hence, if policies affect practices that reduce the amount of pollution generated by an activity, this will not be captured in our analysis.

### 3.2. Calculation of Policy Strength

This paper uses the quantitative method for technological innovation policy proposed by Peng et al. (2008) [44] which has been widely used [45,46,47,48,49], to analyze the strength of policies for controlling ANPS pollution. To improve the accuracy of the scoring criteria (details are put in the Appendix A) and reduce the subjectivity of the scoring, we formed a policy research team with 7 members. To ensure that the evaluations were accurate and reflected an accurate assessment of the policies, the team included 5 researchers who worked at Nanjing Forestry University, Southwest University of Science and Technology, and 2 officers who worked in the Ministry of Agriculture and Rural Affairs of the People’s Republic of China. Each policy was evaluated using the criteria shown in Table 3. For each indicator, if three people give the lowest and four people give the highest score, the team members discussed and revised the quantitative standards of the policy, and vice versa.

This paper comprehensively considers the strength of the policies along four dimensions: policy intensity, policy objectives, policy monitoring, and policy measures. Policy intensity captures the administrative influence of the policy documents for controlling ANPS pollution. The laws database of Peking University includes local laws and regulations documents with six levels but the “local judicial documents” and “administrative licensing approval documents” are less relevant to ANPS pollution. Therefore, the value of policy intensity is from one to four points for the four types of documents: local regulations, local government rules, local normative documents, and local working documents (details of the differences in policies documents are in the Appendix A). Policy objectives are the goals, requirements, and effects expected to be achieved by the implementation of a policy. To be consistent with the policy intensity, these are also scored from one to four points. Policy monitoring captures the response from farmers or other policy audiences to policy implementation in practice. According to the policy documents we have collected, there are fewer clauses on policy monitoring in the content because the government started late in promulgating public environmental participation policies in China. Therefore, we set the value of policy monitoring on a scale of one to three. The Policy measures variables capture the comprehensiveness and feasibility of the methods and tools used to achieve policy goals. Four types of policy measures are considered: administrative regulations, economic incentives, technical support, and educational measures. Policy measures scores are from one to five based on the clarity of the description of the relevant content in the document text.

Firstly, the score of policy intensity, goals, monitoring, and measures of all policy documents of each province (city, autonomous region) in each year are summed up separately. Secondly, the entropy method is used to get a “comprehensive policy strength” index of each province (city, autonomous region) in each year based on the results of the first step. The strength calculation of policy with four types of measures and with only one type of measure are as follows:(2)$$PS_{it}=w_{IN}{\left(IN\right)}^{\prime }{}_{it}+w_{OB}{\left(OB\right)}^{\prime }{}_{it}+w_{FE}{\left(MO\right)}^{\prime }{}_{it}+w_{AMD}{\left(y_{AMD}\right)}^{\prime }it+w_{ECO}{\left(y_{ECO}\right)}^{\prime }it+w_{TEC}{\left(y_{TEC}\right)}^{\prime }it+w_{EDU}{\left(y_{EDU}\right)}^{\prime }it$$
(3)$$ADM_{it}=w_{IN}{\left(IN\right)}^{\prime }{}_{it}+w_{OB}{\left(OB\right)}^{\prime }{}_{it}+w_{FE}{\left(MO\right)}^{\prime }{}_{it}+w_{AMD}{\left(y_{AMD}\right)}^{\prime }it$$
(4)$$ECO_{it}=w_{IN}{\left(IN\right)}^{\prime }{}_{it}+w_{OB}{\left(OB\right)}^{\prime }{}_{it}+w_{FE}{\left(MO\right)}^{\prime }{}_{it}+w_{ECO}{\left(y_{ECO}\right)}^{\prime }it$$
(5)$$TEC_{it}=w_{IN}{\left(IN\right)}^{\prime }{}_{it}+w_{OB}{\left(OB\right)}^{\prime }{}_{it}+w_{FE}{\left(MO\right)}^{\prime }{}_{it}+w_{TEC}{\left(y_{TEC}\right)}^{\prime }it$$
(6)$$EDU_{it}=w_{IN}{\left(IN\right)}^{\prime }{}_{it}+w_{OB}{\left(OB\right)}^{\prime }{}_{it}+w_{FE}{\left(MO\right)}^{\prime }{}_{it}+w_{EDU}{\left(y_{EDU}\right)}^{\prime }it$$
where $PS_{it}$ is the overall strength of the policy document; The variables $ADM_{it}$, $ECO_{it}$, $TEC_{it}$, and $EDU_{it}$ are measures of policy strength with one of four different types of measures: administrative regulations, economic incentives, technical support and the education measures respectively. The variables $w_{IN}$,$\;w_{OB}$, $w_{FE}$ represent the weight of three indicators: policy intensity, policy objective, and policy monitoring calculated by entropy methods; the standardized value which is obtained by using the value in the current year divided by the maximum value: (x’*_it_* = x*_it_*/x*_max_*), ${\left(IN\right)}^{\prime }it$, ${\left(OB\right)}^{\prime }{}_{it}$, ${\left(FE\right)}^{\prime }{}_{it}$, are three indicators of *i*th province in *t*th year. $w_{AMD}$, $w_{ECO}$, $w_{TEC}$, $w_{EDU}$ represent the weight of administrative regulation, economic incentive, technical support, and education measures respectively; ${\left(y_{AMD}\right)}^{\prime }it$, ${\left(y_{ECO}\right)}^{\prime }it$, ${\left(y_{TEC}\right)}^{\prime }it$, ${\left(y_{TEC}\right)}^{\prime }it$ are the standardized value of four policy measures of *i*th province in *t*th year.

### 3.3. Empirical Model and Variable Definitions

The purpose of this study is to examine the impact of environmental policies on ANPS pollution emissions. The fixed effects model and random effects model are two commonly used estimation strategies for panel data. If the Random variable $\mu_{i}\;$representing individual heterogeneity is correlated with an explanatory variable, it is called the Fixed Effects Model (FE); if $\mu_{i}\;$is not correlated with all explanatory variables, it is called Random Effects Model (RE). From the perspective of economic theory, random effects models are rare because, in general, unobserved heterogeneity usually has an impact on explanatory variables [50]. Therefore, the fixed effects model was first used for analysis in this research. Further, the amount of ANPS pollution emissions will not change significantly in a short period and may depend on the number of pollution emissions in the past. Moreover, the effect of environmental policies implemented in the current period has a time lag too. Therefore, we use a dynamic panel data model with the lagged values of ANPS pollution and policy strength. The dynamic panel data models are estimated by using the Generalized Method of Moments (GMM).

The empirical model is set in the following form:(7)$$ANPS_{it}=\alpha_{0}+\alpha_{1}PS_{i,t-1}+\alpha_{2}C_{it}+\delta_{i}+\gamma_{it}$$
(8)$$ANPS_{it}=\alpha +\beta_{1}ANPS_{it-1}+\beta_{2}PS_{it-1}+\rho C_{it}+\mu_{i}+\varepsilon_{it}$$
where $ANPS_{i,t-1}\;\mathrm{and}\;ANPS_{i,t-1}$ denote the ANPS pollution emissions in province *i* in year *t* and with a lag of one year, and $PS_{it-1}$ is the policy strength with a lag of one year. An unobservable random variable, $\mu_{i}$, is a fixed effect that captures province-level heterogeneity; a perturbation term, $\varepsilon_{it}$, varies with time and province; and the vector of control variables, $C_{it}$, includes agricultural production scale, agricultural structure, urbanization rate, and the wealth of rural residents (These data come from China Statistical Yearbook and China Rural Statistical Yearbook). We control for the agricultural production scale that may affect the technology improvements and the chemical resource consumption for agricultural production. Considering that the change in the agricultural structure may affect the proportion of ANPS emissions from different sources, the agricultural structure is added into the regression, which is characterized by the proportion of agricultural production value in the total output value of agriculture, forestry, animal husbandry, and fishery. We control for the urbanization rate, which may increase the chemical fertilizer input due to a lack of labor [51]. The farmers’ wealth is also included in the model to capture the trade-offs farmers face between increasing profit and protecting the environment [52].

As mentioned in the introduction, we allow for the fact that different areas may have different policy effects on ANPS pollution emissions. We divide the data into four regions: eastern, central, western, and northeastern (the location of the regions are presented in a map in the Appendix A) and use the baseline empirical model to estimate the policy effects separately.

In addition, we test whether the effects of different types of measures on ANPS pollution. We divide the policy measures into administrative regulation, economic incentive, technical support, and education measure, and estimate each measure individually.

In the new regressions, we use the strength of each measure to replace the whole policy strength. The model specifications are as follows:(9)$$ANPS_{it}=\alpha +\beta_{1}ANPS_{it-1}+\beta_{2}AMD_{it-1}+\rho C_{it}+\mu_{i}+\varepsilon_{it}$$
(10)$$ANPS_{it}=\alpha +\beta_{1}ANPS_{it-1}+\beta_{2}ECO_{it-1}+\rho C_{it}+\mu_{i}+\varepsilon_{it}$$
(11)$$ANPS_{it}=\alpha +\beta_{1}ANPS_{it-1}+\beta_{2}TEC_{it-1}+\rho C_{it}+\mu_{i}+\varepsilon_{it}$$
(12)$$ANPS_{it}=\alpha +\beta_{1}ANPS_{it-1}+\beta_{2}EDU_{it-1}+\rho C_{it}+\mu_{i}+\varepsilon_{it}$$

The descriptive statistics for all variables are presented in Table 4.

## 4. Results

### 4.1. Construction of TN, TP, and COD Emissions

Among the ANPS pollution emissions in China from 2010 to 2019, the amount of COD emissions is the largest (62.4813 million tons), followed by TN (10.1982 million tons) and TP (1.6475 million tons) emissions (shown in Figure 1). The ANPS might fall due to changes in production. During the period from 2010 to 2019, the fluctuation of TN and TP emissions was relatively balanced, and the changing trend of COD emissions was relatively large during the period from 2015 to 2018. Especially from 2016 to 2017, the total COD discharge in 2016 was 65.2718 million tons while in 2017 was 58.5086 million tons because the final amount of cattle in 2017 decreased by 15.27% compared with the year 2016. The proportion of each specific pollution source (fertilizer source, livestock and poultry breeding source, aquaculture source, straw source, and domestic source) in the TN, TP, and COD emissions are analyzed as shown in Figure 2, Figure 3 and Figure 4. The proportion of each pollution unit in the TN and TP emissions has not changed significantly. In addition to COD emissions, the proportion of rural domestic sewage has increased, which may also be due to the absence of COD discharge from chemical fertilizers.

### 4.2. Calculation Results of Policy Intensity, Objectives, Monitoring, and Measures

In this paper, the classification of policy documents is an indicator to present the policy intensity. Among the 1113 provincial policy documents related to ANPS pollution control, there are 140 local regulations, 15 local government rules, 270 local normative documents, and 688 local working documents. As shown in Figure 5, the average score of policy intensity has a U-shaped change feature, and it has an obvious upward trend from 2017 to 2019, rising from 1.54 in 2017 to 2.44 in 2019. Policy objectives are used to describe the correlation between policy targets and ANPS pollution control. Figure 6 shows the average score of policy objectives (OB) from 2010 to 2019. The average score of policy objectives starts to increase since 2014 which indicates that the objectives of policy documents are more accurate and clear because of the guidance from the policies issued by the central government in 2015–2017. Figure 7 shows the trend of the average score of policy monitoring (MO) from 2010 to 2019. At present, there are few relevant terms of policy monitoring, but the upward trend since 2013 can still be seen. As seen in Figure 8, the average scores of the four different types of measures from 2010 to 2019 were relatively stable. There is substantial variation in the strength across policies with technology support measures being consistently ranked and education measures lowest.

### 4.3. The ANPS Pollution and Policy Strength

According to the calculation results in this paper, ANPS pollution emissions and policy strength in different provinces are quite different. Considering the lag of policy effects, we report the policy strength in 2018 and ANPS pollution emissions in 2019 in 31 provinces of China in Figure 9.

In 2019, Henan, Hebei, and Hunan provinces are provinces with the highest level of ANPS pollution emissions, while Shanghai, Beijing, and the Ningxia Hui Autonomous Region rank the last 3 lowest. Looking at the strength of their policies in 2018, those in Anhui, Fujian, and Hainan were the most stringent and those in Ningxia, Chongqing, and Qinghai were the weakest. Overall, there is no obvious correlation between ANPS pollution emissions and policy strength. In some areas like Beijing and Shanghai City, the proportion of agricultural industry in the GDP is small and the corresponding emissions of ANPS pollution are low, but the policy strength of controlling ANPS pollution control is quite high. The absence of any correlation indicates the importance of using time series data to identify any relationship between policies and ANPS pollution.

### 4.4. Regression Results for Effects of Policy Strength

Table 5 reports the regression results for the effects of policy strength. Firstly, we use a fixed effect model to estimate the impact of current policy strength on ANPS pollution in the same year and the next year. The results show that policies in the current year significantly reduce the emissions of ANPS pollution in the next year. Because of the lag, then, we investigated the impact of policy strength on ANPS pollution by the dynamic panel model with the system GMM method. The *p* values of AR (1) and AR (2) of the dynamic panel models show that there is no second-order autocorrelation problem in the difference of the disturbance term. The results of the Sargan test show that there is no problem of over-identification of instrumental variables.

In the results of the dynamic panel model, the estimated coefficient of ANPS pollution emissions lagging behind by one year are positive and statistically significant at a 1% level, which indicates a strong inter-period correlation of the ANPS emissions. The estimated coefficient of policy strength is negative and statistically significant at the 5% level. A negative coefficient implies that the stronger the policy strength in the current period, the lower the ANPS pollution will be in the next period. However, the small absolute value of the estimated coefficient on policy strength indicates that the impact of policies is still limited. We find that there is a positive and statistically significant association between GDP and ANPS pollution, which implies that as the GDP increases, the emissions of ANPS pollution increase too. The estimated coefficient of the urbanization rate is significantly negative at the 1% level. This result is not as the expectation mentioned above in Section 3.3 but consistent with findings by Ma and Wang (2021) [53] and Yan et al. (2022) [54]. Recalling the process of urbanization in China, it can be inferred that reducing of rural labor force improves the degree of agricultural mechanization and environment-friendly technology for agricultural production, which can reduce ANPS pollution.

The relationship between policy strength and ANPS pollution emissions is not robust across regions (Table 6). We still use the dynamic panel model to estimate policy effects on ANPS pollution in four regions of China separately: eastern, central, western, and northeastern. While the estimated coefficients of policy strength are negative and statistically significant in the northeastern region, there is no statistically significant correlation for the other regions. Some of the coefficients on the other control variables merit brief comments. We see that the level of urbanization rate has a significant negative effect on ANPS pollution in central and western China. The urbanization rate has a positive effect on ANPS pollution in Northeastern China but is not statistically significant.

### 4.5. Regression Results for Effects of Policy with Different Measures

As shown in Table 7, the statistically significant effect of policy strength at the provincial level is robust when each of the measure types is considered independently. We find that the negative effect of policy measures on ANPS pollution emissions is statistically significant at the 5 or 10% level for each measure.

The results indicate that different types of measures can impact ANPS pollution. Administrative regulations are often used to restrict farmers’ chemical input by words “prohibition”, “not allowed”, “restriction”, and “standardization”, which can lead to a reduction of the emissions of ANPS pollution. Economic incentive measures seek to internalize external costs by using subsidies, taxes, rewards, and fines, which could remind farmers of the environmental costs and encourage them to change production behavior in an environmentally friendly way. Technical support measures seek to help farmers understand the details of ANPS-reducing technologies or how to construct new environmental protection facilities that can decrease ANPS pollution emissions. Education measures involve explanations and instructions on environmental protection and technologies for agricultural production that can help to increase farmers’ environmental protection awareness and improve agricultural technology that is correlated with ANPS pollution [55,56,57,58]. Table 7 suggests that all of these types of measures have had a small but statistically significant impact on ANPS pollution in China.

### 4.6. Robustness Check

In this paper, we are interested in whether there is a negative correlation between official policies and ANPS pollution. In this section, we discuss four robustness tests of the estimated results for policy effects. Firstly, we replace the measurement method of the explained variable, the emissions of ANPS pollution. The entropy method has been used to process the usage data of fertilizer, pesticide, agricultural plastic film, and agricultural diesel oil. Secondly, we calculate the core explanatory variable “policy strength” using the Principal Component Analysis (PCA) method, a dimensionality-reducing process as an alternative to the entropy method. Thirdly, we use the differential GMM method to estimate the dynamic panel model, which can eliminate omitted variable bias due to unobserved cross-sectional individual effects.

As shown in Table 8, the coefficient of the key variables, lagged ANPS pollution and Policy Strength, have the same sign and similar significance levels to those in the estimation of the dynamic panel model with the system GMM method. In this case, the results are very similar to the results in Table 5, suggesting that our results are robust to alternative specifications.

Moreover, the *p*-values of AR (2) and the Sargan test are all greater than 0.05, which means that the null hypotheses of “the second-order autocorrelation coefficient of the difference of the disturbance term are 0” and “all instrumental variables are valid” are accepted, indicating the applicability of the dynamic panel data model. The lag of the 1-year term of comprehensive policy strength is significantly negative in the two models, and its absolute value is relatively close, which is basically consistent with the estimation results above.

## 5. Discussion

The results above provide some evidence that government policies have reduced ANPS pollution in China—as a policy becomes more stringent in emissions in the following year’s fall. Examples of long-established and largely successful regulations include the Clean Water Act in the USA, the Water Framework Directive in European Union, and the National Ecological Environment Protection Outline in China. In 1972, the United States promulgated the Clean Water Act (CWA). After more than 50 years of implementation, it has effectively reduced water pollution and improved water quality [59]. The experience of the United States also reflects the experience of most countries in applying policies for controlling pollution [60].

Command and control type policies such as the administrative rules studied in this analysis can reduce ANPS pollution by setting official standards of environmental quality and protection rules. So long as the standards and rules are understood and respected by farmers, agricultural production behavior should change, thus alleviating the emissions of ANPS pollution. However, this is not easy to achieve. Hence, different types of measures are used to promote the effectiveness of policy objectives. According to the regression results of this paper, each measure is conducive to the control of ANPS pollution.

Regionally, the regression results of this paper indicate that policies for controlling ANPS pollution only have a statistically significant effect in the northeastern area. There are regional differences in the effect of policies on pollution reduction. Other scholars have reached a similar conclusion. Zhang et al. (2018) [61] found that the marginal utility of environmental regulation on green economic efficiency is most obvious in the eastern region by analyzing China’s provincial statistics data from 2000 to 2015. Wang et al. (2018) [62] showed that environmental policy had a stronger effect on carbon emissions in the eastern and central regions. The northeastern region includes Heilongjiang, Jilin, and Liaoning provinces, which rank in the top five in terms of agricultural GDP and are the main grain production areas in China. Under the pressure of economic development and environmental protection, the government will need to pay more attention to agricultural environment protection, and therefore show more power of policies to reduce pollution emissions.

Among the control variables, we find consistent evidence of a positive relationship between ANPS pollution and agricultural GDP and a negative relationship with the urbanization rate. Urbanization is accompanied by rural labor emigration. Some scholars have argued that as surplus rural labor falls, farmers will make up for the resulting labor shortage by increasing other inputs such as chemical fertilizers and pesticides, aggravating ANPS pollution (Jiang et al., 2021). The results of this paper show that the increases in the urbanization rate lead to reductions in ANPS pollution. A possible reason is that the reduction of rural labor can force or promote the improvement of agricultural production technology which can reduce ANPS pollution through higher production efficiency.

## 6. Conclusions

Finding ways to reduce ANPS pollution is critical if China’s agriculture sector is to develop in a sustainable way. In this paper, we estimate ANPS pollution emissions from five sectors from 2010 to 2019. We also develop indicators of policy strength for 31 provinces and a total of 1113 policy documents. These new data sets allow us to study the relationships between ANPS pollution and policy strength in different provinces and across time. Henan, Hebei, and Hunan provinces are the top 3 areas with the highest ANPS pollution emissions in 2019, while Ningxia Hui Autonomous Region, Chongqing city, and Qinghai province are ranked the lowest three places in terms of policy strength in 2018. According to the details and frequency, the average strength of technical support measures is the highest of the four types over the studied period.

We build a dynamic panel model by using systematic GMM methods to investigate the relationship between policies and ANPS pollution emissions in China. The regression results show that there is a small but statistically significant negative association between policy strength and ANPS pollution, and this result is robust when we restrict our regression to different types of policy measures. When we break down our analysis by region, we find evidence that policy strength has a statistically significant effect only in the northeastern area of China. When we analyze the effects of the four types of policy measures, it appears that all contribute to the reduction in ANPS pollution.

There are limitations of the current analysis that suggest the need for further work in this area. First, due to data limitations, we hold constant the loading coefficients when calculating ANPS pollution. If policies have the effect of reducing these coefficients, then our estimates of the impact of policy on ANPS emissions will be biased downward in absolute value. Second, the mechanism of action of various types of policy measures is also worth further exploration. Third, when we made the scoring criteria for the calculation of policy strength, the research team only included 2 officers who worked in the Ministry of Agriculture and Rural Affairs of the People’s Republic of China. More officers on the team might improve the validity of our measures of the effectiveness of policy strength in future research. Fourth, we don’t compare these policy measures and describe which one is more effective. Fifth, while we have explored the data using what we believe are the most appropriate statistical methods, there are certainly other specifications that might be explored. Hence, the data are available for asking so that other researchers might carry out their own analysis.

Overall, our study suggests that China’s policies have been helpful in controlling ANPS pollution though there are important regional differences. Although the government can use administrative regulations that impose strict limits on agricultural chemical inputs or can provide technical support programs to reduce ANPS pollution, the cost of these approaches is probably high. We find that economic incentives and educational measures can be effective for reducing ANPS pollution from the regression results of different policy measures, while the average strength of policy with economic incentives and educational measures from 2010 to 2019 is lower than the other two measures because of less content and details. Hence, this suggests that the Chinese government should take advantage of economic incentives and education measures to change farmers’ behavior to achieve the goal of addressing the important problem of ANPS pollution.

## Figures and Tables

**Figure 1 ijerph-20-03741-f001:**
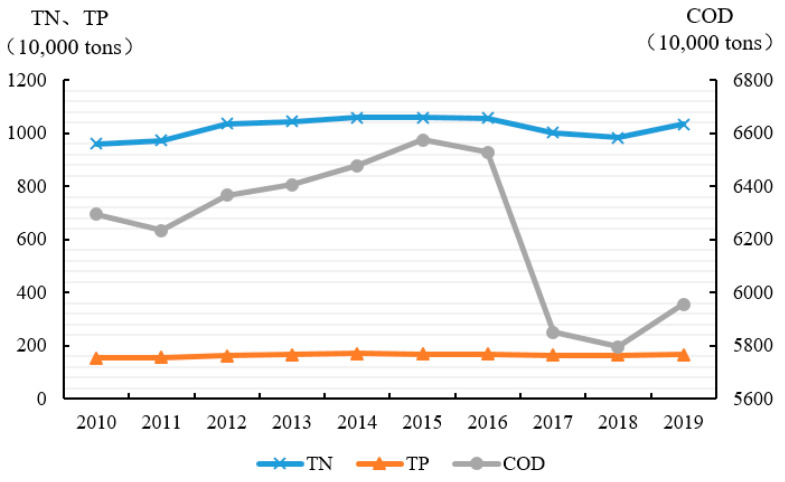
Average amount of TN, TP, and COD emissions from 2010 to 2019.

**Figure 2 ijerph-20-03741-f002:**
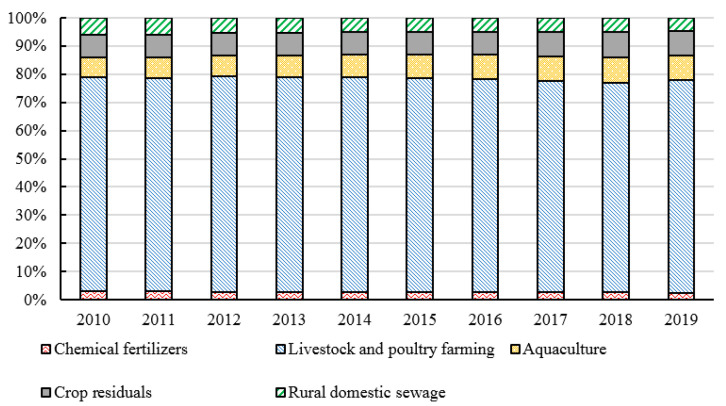
The proportion of TN emissions from different sources from 2010 to 2019.

**Figure 3 ijerph-20-03741-f003:**
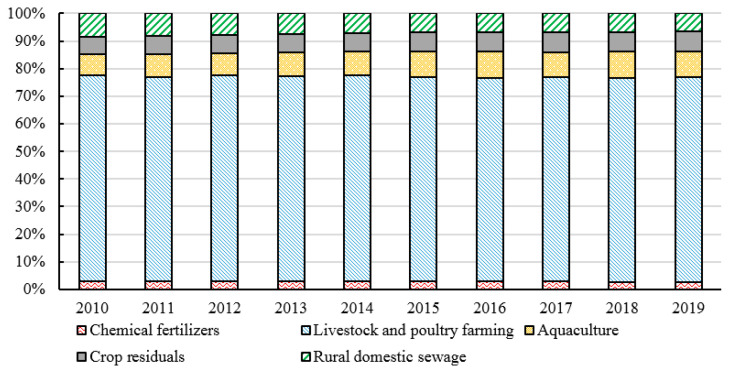
The proportion of TP emissions from different sources from 2010 to 2019.

**Figure 4 ijerph-20-03741-f004:**
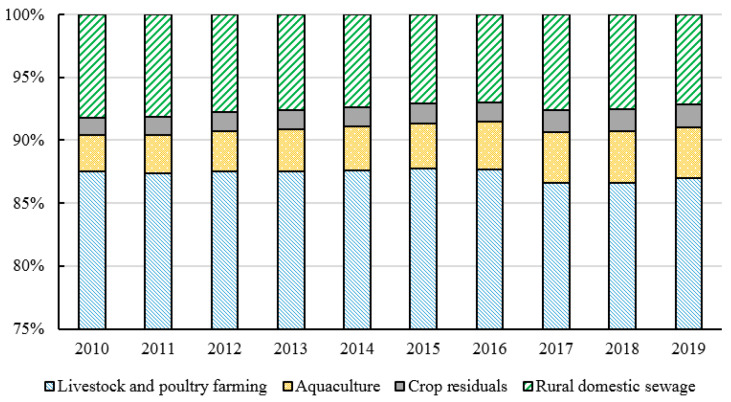
The proportion of COD emissions from different sources from 2010 to 2019.

**Figure 5 ijerph-20-03741-f005:**
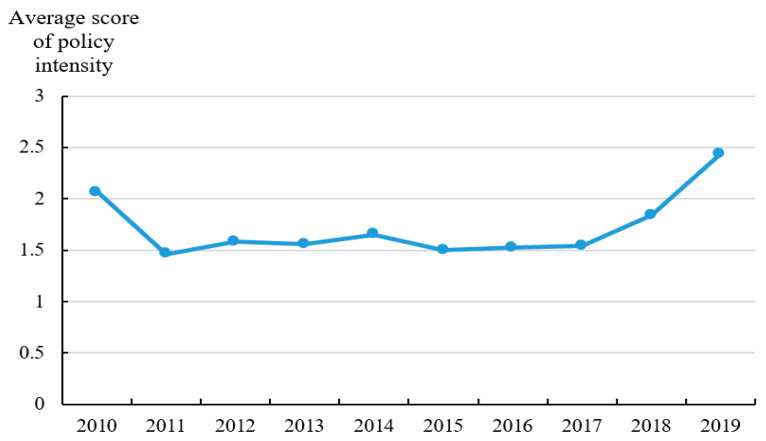
Average score Policy intensity from 2010 to 2019.

**Figure 6 ijerph-20-03741-f006:**
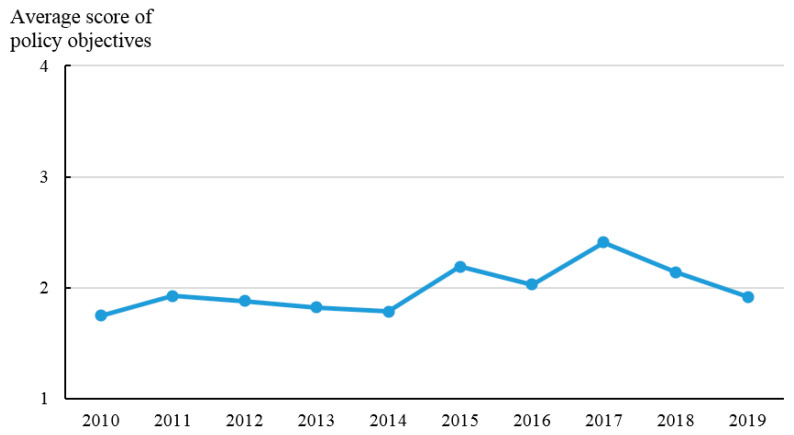
Average score Policy objectives from 2010 to 2019.

**Figure 7 ijerph-20-03741-f007:**
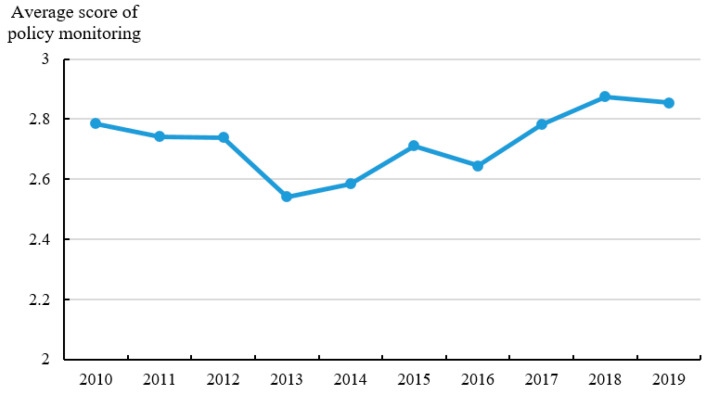
Average score of policy monitoring from 2010 to 2019.

**Figure 8 ijerph-20-03741-f008:**
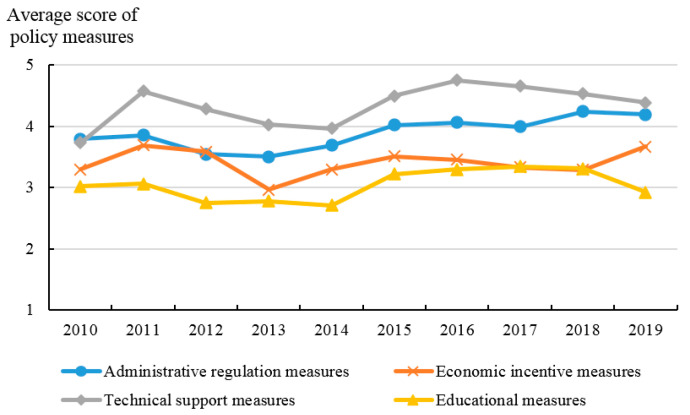
Average strength of policy with four types of measures from 2010 to 2019.

**Figure 9 ijerph-20-03741-f009:**
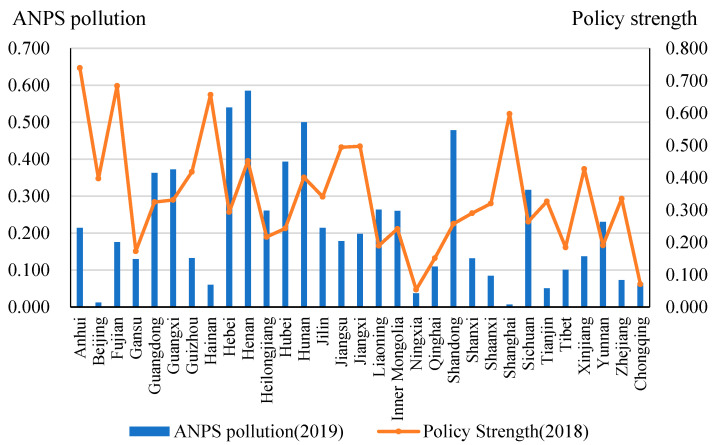
Policy strength in 2018 and ANPS pollution emissions in 2019.

**Table 1 ijerph-20-03741-t001:** Agricultural Non-point Source Pollution Measurements.

Source	Category	Unit	Key Variables
Planting	Chemical fertilizer	Nitrogen fertilizer, Phosphate fertilizer, Compound fertilizer	Total input amount of kth fertilizer (k = 1, 2, 3)
	Crop residue	Crop residue of rice, wheat, corn, beans, potatoes, oilseeds, and vegetables	Total amount of *m*th crop yield (m = 1, 2, …, 7)
Animal breeding industry	Livestock and poultry farming	Pig, cattle, poultry	Slaughter quantity of pigs (z1) and poultry (z2), Stock quantity of cattle (z3)
	Aquaculture	Marine and freshwater aquaculture	production of *r*th aquaculture (r = 1, 2)
Rural community	Rural domestic sewage		Rural population (pop)

**Table 2 ijerph-20-03741-t002:** Equations for Calculating ANPS Pollution.

Chemical Fertilizer		Crop Residue	
TN	$F_{n}=\overset{3}{\underset{k=1}{\sum }}T_{k}{*\lambda }_{{\mathrm{kF}}_{n}}$	TN	$S_{n}=\overset{7}{\underset{m=1}{\sum }}T_{m}{*\varphi }_{m}{*\lambda }_{S_{n}}$
TP	$F_{p}=\overset{3}{\underset{k=1}{\sum }}T_{k}{*\lambda }_{{\mathrm{kF}}_{p}}$	TP	$S_{p}=\overset{7}{\underset{m=1}{\sum }}T_{m}{*\varphi }_{m}{*\lambda }_{S_{p}}$
COD	—	COD	$S_{\mathrm{cod}}=\overset{7}{\underset{m=1}{\sum }}T_{m}{*\varphi }_{m}{*\lambda }_{S_{c}}$
Livestock and poultry farming		Rural domestic sewage	
TN	$L_{n}=\overset{3}{\underset{z=1}{\sum }}T_{z}{*\theta }_{\mathrm{zL}}{*\lambda }_{{\mathrm{zL}}_{n}}$	TN	$E_{n}={\mathrm{pop}*\lambda }_{E_{n}}$
TP	$L_{p}=\overset{3}{\underset{z=1}{\sum }}T_{z}{*\theta }_{\mathrm{zL}}{*\lambda }_{{\mathrm{zL}}_{p}}$	TP	$E_{p}={\mathrm{pop}*\lambda }_{E_{p}}$
COD	$L_{\mathrm{cod}}=\overset{3}{\underset{z=1}{\sum }}T_{z}{*\theta }_{\mathrm{zL}}{*\lambda }_{{\mathrm{zL}}_{c}}$	COD	$E_{\mathrm{cop}}={\mathrm{pop}*\lambda }_{E_{c}}$
Aquaculture			
TN	$A_{n}=\overset{2}{\underset{r=1}{\sum }}T_{r}{*\lambda }_{A_{n}}$		
TP	$A_{p}=\overset{2}{\underset{r=1}{\sum }}T_{r}{*\lambda }_{A_{p}}$		
COD	$A_{\mathrm{cod}}=\overset{2}{\underset{r=1}{\sum }}T_{r}{*\lambda }_{A_{c}}$		

**Table 3 ijerph-20-03741-t003:** Indicators of Policy Strength.

Indicator	Description	Value
Policy intensity (*IN*)	The administrative influence of policy documents	1–4
Policy objectives (*OB*)	The goals, requirements, and effects expected to be achieved by the implementation of a policy	1–4
Policy monitoring (*MO*)	The response from farmers or other policy audiences to policy implementation in practice.	1–3
Indicators of Policy measures		
$\mathrm{Administrative}\;\mathrm{regulation}\;\left(y_{AMD}\right)$	The official regulations include a maximum number of chemical inputs, standards for chemical inputs use, constraints, and restrictions on polluting behavior.	1–5
$\mathrm{Economic}\;\mathrm{incentive}\;\left(y_{ECO}\right)$	The awards, fines, taxes, and subsidies encourage farmers’ environmentally friendly behavior.	1–5
$\mathrm{Technical}\;\mathrm{support}\;\left(y_{TEC}\right)$	The methods or technologies can be used in the process of agricultural activities for reducing ANPS pollution	1–5
$\mathrm{Educational}\;\left(y_{EDU}\right)$	The information on agricultural technical training courses, technical guidance on agricultural production, methods of using chemical inputs and agricultural tools, and so on.	1–5

**Table 4 ijerph-20-03741-t004:** Description of major variables.

Variable	Mean	Std. Dev
ANPS pollution emissions (ANPS, scale: 0.0–1.0)	0.2188	0.1641
Policy strength (PS, scale: 0.0–1.0)	0.2166	0.1789
Strength of Administrative regulation measures (AMD scale: 0.0–1.0)	0.2155	0.1785
Strength of economic incentive measures (ECO, scale: 0.0–1.0)	0.2202	0.1791
Strength of technical support measures (TEC, scale: 0.0–1.0)	0.2121	0.1757
Strength of education measures (EDU, scale: 0.0–1.0)	0.2123	0.1775
Economic scale in the agricultural sector (log of GDP in hundred million yuan)	7.4630	1.0789
Agricultural structure (AS, scale: 0.0–1.0)	0.5244	0.0844
Urbanization rate (URB, scale: 0.0–1.0)	0.5609	0.1339
Wealth of rural residents (log of INC in yuan per person)	8.9034	0.3097

**Table 5 ijerph-20-03741-t005:** Regression results of the impact of policy strength on ANPS pollution.

Variable	Fixed Effect Model	Fixed Effect Model (With Lagged Policy Strength)	Dynamic Panel Model (With Lagged Policy Strength)
Lag ANPS			0.7767 *** (0.0882)
Policy Strength	−0.0017 (0.0051)		
Lag Policy Strength		−0.0101 * (0.0058)	−0.0136 ** (0.0063)
GDP	0.0774 ** (0.0295)	0.0775 ** (0.0305)	0.0378 * (0.0196)
Agricultural Scale	−0.1561 ** (0.0680)	−0.1357 ** (0.0661)	−0.0308 (0.0725)
Urbanization rate	−0.2416 (0.2039)	−0.2921 (0.2250)	−0.2583 *** (0.0895)
Wealth of rural residents	0.0082 (0.0635)	0.0308 (0.0755)	0.0581 (0.0545)
Constant	−0.2137 (0.5843)	−0.3964 (0.6778)	−0.5841 (0.4117)
N	310	310	279
F	2.41	3.23	
R2	0.5856	0.5891	
AR (1)			0.0756
AR (2)			0.0549
Sargan			0.1436

Note: ***, **, * Represent significance levels of 1%, 5%, and 10%, respectively, with robust standard errors in parentheses.

**Table 6 ijerph-20-03741-t006:** Regression results of policy effects on ANPS pollution in different areas.

Variable	Eastern	Central	Western	Northeastern
Lag ANPS	0.9455 *** (0.0797)	1.0092 *** (0.0588)	0.9401 *** (0.1428)	0.6384 *** (0.0788)
Lag Policy Strength	−0.0015 (0.0081)	0.0086 (0.0181)	−0.0056 (0.0134)	−0.0861 ** (0.0409)
GDP	0.0013 (0.0217)	−0.1334 (0.0969)	0.0028 (0.0423)	−0.0048 (0.0561)
Agricultural Scale	−0.1409 (0.1501)	0.8095 (0.5766)	0.0336 (0.0613)	0.1333 (0.1298)
Urbanization rate	−0.1943 (0.1774)	−0.6382 * (0.3757)	−0.2472 * (0.1438)	0.2504 (0.4007)
Wealth of rural residents	−0.0004 (0.1174)	0.6570 (0.4483)	0.1166 (0.0954)	0.0384 (0.0424)
Constant	0.2077 (1.1029)	−4.8441 (3.2669)	−0.9121 (0.6274)	−0.1870 (0.3719)
N	90	54	108	27
AR (1)	0.0634	0.1514	0.1579	0.0929
AR (2)	0.4396	0.3467	0.7953	0.2585
Sargan	0.9971	0.1074	0.9608	0.7232

Note: ***, **, * Represent significance levels of 1%, 5%, and 10%, respectively, with robust standard errors in parentheses.

**Table 7 ijerph-20-03741-t007:** Regression results of effects of policy with different measures on ANPS pollution.

Variables	Admin. Measures	Economic Measures	Technical Support	Educational Measures
L.ANPS	0.7796 *** (0.0908)	0.7786 *** (0.0901)	0.7791 *** (0.0905)	0.7776 *** (0.0902)
Lagged Admin. Measures strength	−0.0120 * (0.0067)			
Lagged Economic Measures strength		−0.0125 ** (0.0062)		
Lagged Technical Support strength			−0.0127 * (0.0066)	
Lagged Educational Policies strength				−0.0126 * (0.0066)
GDP	0.0362 * (0.0189)	0.0368 * (0.0193)	0.0365 * (0.0191)	0.0367 * (0.0190)
Agricultural Scale	−0.0299 (0.0713)	−0.0292 (0.0707)	−0.0289 (0.0701)	−0.0297 (0.0721)
Urbanization rate	−0.2568 *** (0.0927)	−0.2594 *** (0.0920)	−0.2578 *** (0.0924)	−0.2583 *** (0.0924)
Wealth of rural residents	0.0566 (0.0522)	0.0577 (0.0523)	0.0573 (0.0525)	0.0576 (0.0518)
Constant	−0.5602 (0.4043)	−0.5740 (0.4013)	−0.5689 (0.4053)	−0.5724 (0.4001)
N	279	279	279	279
AR (1)	0.0769	0.0769	0.0768	0.0778
AR (2)	0.0549	0.0501	0.0555	0.0533
Sargan	0.1513	0.1490	0.1505	0.1505

Note: ***, **, * represents 1%, 5%, and 10% significant level respectively, robust standard errors are illustrated in parentheses.

**Table 8 ijerph-20-03741-t008:** Robustness test results of policy effects on ANPS pollution.

Variables	ANPS Pollution by Entropy Method	Policy Strength by Principal Component Analysis	Differential GMM Model
Lag. ANPS	1.0262 *** (0.0474)	0.7766 *** (0.0880)	0.5352 *** (0.0463)
Lag. Policy Stringency	−0.0087 * (0.0046)	−0.0010 ** (0.0004)	−0.0146 ** (0.0067)
Control variables	YES	YES	YES
Constant	−0.1967 (0.3338)	−0.5875 (0.4142)	−1.0864 ** (0.5248)
N	279	279	279
AR (1)	0.0367	0.0753	0.0666
AR (2)	0.5177	0.0557	0.0573
Sargan	0.2493	0.1428	0.0981

Note: ***, **, and * represent significance levels of 1%, 5%, and 10%, respectively, and robust standard errors are in brackets.

## Data Availability

Data can be requested from the authors.

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
