# Peer review of "Effects of Policy for Controlling Agricultural Non-Point Source Pollution in China: From a Perspective of Regional and Policy Measures Differences"

_ijerph, 2023, doi:10.3390/ijerph20043741_

Round 1
Reviewer 1 Report
1) The paper uses quantitative approach, however, the validation from the ministry people should be increased, or at least as focused groups (otherwise, put in limitations)
2) The authors should also mention other driving factors apart from the policy measures from the government. There could be some geographical factors, cultural factors, population density factor, climate, and weather related factors, or even technological advancement factors for ANPS.
3) The results should be validated with several loading factors, otherwise results will be biased.
4) The conclusion or policy suggestion that economic incentives and education incentives are not directly coming from the results that were made. Need a clear result-based recommendation.
5) Without making a cost comparison of the 4 types of interventions, it can't be decisive which one will be effective.
Reviewer 2 Report
The article brings up an issue of great interest that has created a great contradiction, namely pollution generated by agriculture versus agricultural production.
Government interventions are known to be a key factor for pollution control by issuing official documents, and the present study, based on statistical data from 31 provinces from 2010 to 2019, shows that China's policies have been helpful for pollution control, although they have there were important regional differences.
The authors used the entropy method as a working method to calculate the amount of pollution emissions and the strength of the policy, stating that there are few studies that analyze the policies used for pollution control and their effectiveness on pollution.
The elements presented in the form of tables and figures are appropriate to the analyzed topic.
Pay attention to the writing of the article! Authors are invited to study the MDPI recommendations for writing articles as there are several aspects that need to be corrected:
- the citation model requested by the journal is not followed, nor is the model for naming figures,
- footnotes are also used, which are only recommended for tables and figures.
Author Response
Thanks for your comments. The response to your comments are as follows:
Point 1: the citation model requested by the journal is not followed, nor is the model for naming figures,
Response 1: The format of Figure naming and the citation model have been changed according to the author guidance.
Point 2: footnotes are also used, which are only recommended for tables and figures.
Response 2: Considered the limitations of space in the paper, we put some definitions or explanations by using footnotes with small font size in order to provide a better understanding to the readers. There are 7 footnotes in the paper now. We try to shorten the definition and move the sentence to the paper, then delete footnotes from 7 to 3. We removed the footnotes for the loading coefficients, the Peking University Law database, revision of scoring criteria, and standardized value of the indicators.

Reviewer 3 Report
This article uses the entropy method to calculate the emission amount and policy power of Agricultural Non-Point Source (ANPS) pollution of 31 provinces in China from 2010 to 2019. In the study, the system generalized moment dynamic panel data model is used to estimate the effects of policies with different measures on ANPS pollution emissions. According to the findings of the study, China's policies have been helpful in controlling ANPS pollution, although there are significant regional differences. It also concludes that all four types of policy measures contribute to reducing ANPS pollution.
The topic of the article is interesting and current. The work is generally well written. The contributions of the study to the literature are clearly stated. The study has three main contributions to the literature. The authors' first contribution is that we measured the strength of policies implemented in 31 provinces of China from 2010 to 2019. As a second contribution, the authors analyze how well ANPS pollution is controlled by different measures across policies, including administrative regulations, economic incentives, technical supports, and educational efforts. Third, the study examines how policies affect ANPS pollution emissions in 96 different geographic regions.
The variables used in the study are explained in detail. The data and methods used in the calculation of the variables are presented to the reader. The models used in the study are suitable for their purpose. Numerous model predictions have been made. The results of different models are given comparatively. Robustness test results are included for the estimated models. The limitations of the study are clearly stated. I found the bibliography of the study sufficient and up-to-date. The findings of the study were interpreted in detail. I found the writing of the study successful, so I do not request changes from the authors.
Author Response
Thanks very much for your comments and encouragement.
Reviewer 4 Report
Dear Authors,
The presented study concerns the current issue of emissions from agriculture. It is well described and presented.
I just have a few minor comments.
In the abstract, it should be emphasized that the analyzed period is short and the potential impact of changes in the law on the level of emissions cannot be fully verified.
There are some significant weaknesses that should be better described. Is the assessment of the importance of legal regulations based on the opinion of only 7 specialists? Is this a representative sample?
The level of emissions depends on the volume of production. Progress should therefore be measured by emissions per unit of production. Currently, the assessment may result primarily from the change in the size of production and its structure. This is shown by the importance of the "Agricultural scale" variable in some models. It is worth explaining this in the methodology.
The results indicate that the level of emissions depends on the production volume, and the production volume has changed (295-299). It is worth giving the reason for such a change because it is the main reason for changes in emissions.
In Fig. 1, it is better not to cut the right axis at 5400.
Taking into account the results of the models, it should be emphasized that the level of emissions is stable and in the short term, changes (including legal changes) have only a weak impact on emissions and are revealed as significant only in some models. (tab. 6).
It should also be explained why emissions per unit of production are not taken into account - Some sources (Bennetzen, E. H., Smith, P., Porter, J. R. (2016). Agricultural Production and Greenhouse Gas Emissions From World Regions – The Major Trends Over 40 Years. Global Environmental Change, no. 37, pp. 43–55. DOI: 10.1016/j.gloenvcha.2015.12.004
Bennetzen, E.H., Smith, P., Porter, J. R. (2016). Decoupling of Greenhouse Gas Emissions From Global Agricultural Production: 1970–2050. Global Change Biology, vol. 22, no. 2, pp. 763–781. DOI: 10.1111/gcb.13120
Gregory, P., Ingram, J., Andersson, R., Betts, R., Brovkin, V., Chase, T., Grace, P., Gray, A., Hamilton, N., Hardy, T., Howden, S., Jenkins, A., Meybeck, M., Olsson, M., Ortiz-Monasterio, I., Palm, C., Payn, T., Rummukainen, M., Schulze, R., Thiem, M., Valentin, C., Wilkinson, M. (2002). Environmental Consequences of Alternative Practices for Intensifying Crop Production. Agric. Ecosyst. Environ., vol. 88, pp. 279–290. DOI: 10.1016/S0167-8809(01)00263-8)
There are some typos, e.g. in line 156, 157, 162, it's hard to check.
Reviewer 5 Report
This work describes a very important and current problem of environmental pollution associated with ANPS. In my opinion, the possibility of publishing it should be considered, if only because of the subject it raises. This work has been written very carefully, which I perceive positively. It is an interesting theoretical-analytical-expert study, which was based on large data sets with various levels of measurability and detail.
The introduction has been written very skilfully, because it already indicates that this work was analyzed taking into account such defined and measurable data as, for example, COD, TN, TP, as well as a review of documents. As a result, the assumptions made, the calculations carried out and the conclusions drawn become understandable. It is very important that the authors indicate the purposefulness of the undertaken task, but also indicate the uncertainties in the research model they propose. However, I have a doubt. Is such a far-reaching statement (lines 44 to 46) based on only one publication legitimate? Please think it over.
In subchapter 3.2. (line 185) stated that a quantitative method was used to calculate the strength of the policy. In my opinion, however, it was a quantitative and expert method, which should be noted.
In addition, in my opinion, in the methodological part of the work, the location of the regions (eastern, central, western and north-eastern) that were the source of input data for the analysis should be presented graphically (map).
In line 298 there is a sentence which is an unnecessary repetition of the previous sentence.
Why in the title of subchapter 4.2. "policy monitoring" omitted?
I also have a doubt about subchapter 4.5. Are "technical support measures" and "educational measure" not difficult to distinguish in the light of the descriptions presented in this paper. If (line 391) "Technical support measures seek to help farmers understand how ..." then are they not an "educational measure"?
The list of literature must be prepared in accordance with the publisher's guidelines. In many places dots after abbreviations of authors' names are missing, necessary spaces are missing, double dashes appear, etc. In general, this part of the work is rather carelessly prepared.
